# 11-Year Trend in Antibiotic Consumption in a South-Eastern European Country; the Situation in Albania and the Implications for the Future

**DOI:** 10.3390/antibiotics12050882

**Published:** 2023-05-09

**Authors:** Iris Hoxha, Brian Godman, Admir Malaj, Johanna C. Meyer

**Affiliations:** 1Department of Pharmacy, Faculty of Medicine, University of Medicine Tirana, 1001 Tirana, Albania; 2Department of Pharmacoepidemiology, Strathclyde Institute of Pharmacy and Biomedical Sciences, University of Strathclyde, Glasgow G4 0RE, UK; 3Department of Public Health Pharmacy and Management, School of Pharmacy, Sefako Makgatho Health Sciences University, Ga-Rankuwa 0208, South Africa; 4Independent Researcher, 1001 Tirana, Albania; 5South African Vaccination and Immunisation Centre, Sefako Makgatho Health Sciences University, Molotlegi Street, Garankuwa, Pretoria 0208, South Africa

**Keywords:** Albania, antibiotics, antimicrobial resistance, antimicrobial stewardship programs, AWaRe classification, drug utilization, healthcare policy, healthcare professional education

## Abstract

There are growing concerns with rising antimicrobial resistance (AMR) across countries. These concerns are enhanced by the increasing and inappropriate utilization of ‘Watch’ antibiotics with their greater resistance potential, AMR is further exacerbated by the increasing use of antibiotics to treat patients with COVID-19 despite little evidence of bacterial infections. Currently, little is known about antibiotic utilization patterns in Albania in recent years, including the pandemic years, the influence of an ageing population, as well as increasing GDP and greater healthcare governance. Consequently, total utilization patterns in the country were tracked from 2011 to 2021 alongside key indicators. Key indicators included total utilization as well as changes in the use of ‘Watch’ antibiotics. Antibiotic consumption fell from 27.4 DIDs (defined daily doses per 1000 inhabitants per day) in 2011 to 18.8 DIDs in 2019, which was assisted by an ageing population and improved infrastructures. However, there was an appreciable increase in the use of ‘Watch’ antibiotics during the study period. Their utilization rose from 10% of the total utilization among the top 10 most utilized antibiotics (DID basis) in 2011 to 70% by 2019. Antibiotic utilization subsequently rose after the pandemic to 25.1 DIDs in 2021, reversing previous downward trends. Alongside this, there was increasing use of ‘Watch’ antibiotics, which accounted for 82% (DID basis) of the top 10 antibiotics in 2021. In conclusion, educational activities and antimicrobial stewardship programs are urgently needed in Albania to reduce inappropriate utilization, including ‘Watch’ antibiotics, and hence AMR.

## 1. Introduction

There are concerns with rising rates of antimicrobial resistance (AMR) globally, increasing morbidity, mortality, and costs [1,2,3,4,5,6]. In 2019, there were an estimated 1.27 million deaths globally directly attributable to AMR with 4.95 million associated with bacterial AMR and rising [7]. As a result of this alarming trend, AMR is now often seen as the next pandemic [8]. Multiple international, regional, and national activities have been instigated in recent years to reduce AMR. These include the instigation of the World Health Organization (WHO) Global Action Plan to reduce AMR, translated into National Action Plans (NAPs) across a number of countries and continents [1,9,10,11,12,13]. Other key activities driven by the WHO include the instigation of the AWaRe classification, which divides antibiotics into ‘Access’, ‘Watch’, and ‘Reserve’ groups depending on their resistance potential, with greater resistance potential for antibiotics included in the ‘Watch’ and ‘Reserve’ groups, as well as more recently, prescribing guidance [14,15,16]. The initial target to reduce AMR is that at least 60% of antibiotic utilization within a sector or country should be ‘Access’ antibiotics, with ‘Watch’ and ‘Reserve’ antibiotics limited in their utilization [17,18,19]. This is a key issue among low- and middle-income countries (LMICs), with their increasing utilization of ‘Watch’ antibiotics driving up AMR [18,19].

We have seen a number of Central and Eastern European (CEE) countries, as well as former Soviet Union Republics, which are not included in CEE countries, introduce or plan to introduce a number of initiatives and activities to reduce inappropriate utilization of antibiotics and ultimately AMR [20,21,22,23,24,25,26]. In Armenia, total antibiotic consumption decreased from 15.9 defined daily doses (DDDs) per 1000 inhabitants per day (DIDs) in 2011 to 10.7 DIDs in 2015 through targeted awareness campaigns and other initiatives by a multi-sectorial group at the Ministry of Health [23]. In Slovenia, multiple activities, which included prescribing restrictions via the Ministry of Health and the Health Insurance Agency, resulted in a 31% decrease in antibiotic consumption between 1999 and 2012 [20]. Such activities and dynamics contrast with the documentation of increased consumption of antibiotics in Serbia in recent years. Antibiotic utilization rose from 26.4 DIDs in 2011 to 36.5 DIDs in 2015 with limited demand-side measures introduced by the Government of Health Insurance Agency to influence physician prescribing [23]. Similarly in Poland, a lack of demand-side measures among key stakeholder groups helped to maintain continued high antibiotic utilization rates [27]. This included appreciable prescribing of broad-spectrum antibiotics, thereby enhancing resistance rates [27]. Overall, cultural and other factors, including resistance to change alongside regulations, play a key role in addition to other governance factors, with influencing antibiotic prescribing and dispensing patterns [28,29,30,31,32,33]. This includes issues of corruption surrounding governance in healthcare across Europe with poor governance increasing inappropriate antibiotic use and hence AMR [34,35].

Albania is an upper middle-income country in South-East Europe with a population of just under three million in recent years [36]. However, the population in Albania is ageing, which will impact on future medicine use as well as the adequacy of healthcare funding. Good governance in a country is important to tackle AMR [34]. This is a concern in Albania, with Albania currently ranked 101 out of 180 countries in terms of perceptions of corruption [37]. There are also concerns regarding current knowledge of key stakeholder groups in Albania regarding antibiotics and AMR [38,39]. Alongside this, there are concerns regarding the current lack of antimicrobial stewardship (AMS) activities in Albania to influence future utilization. In addition, despite current legislation prohibiting the purchasing of antibiotics without a prescription, this still occurs, exacerbated by patient pressure and weak governance [34,38,40,41,42]. We have seen in other ex-communist countries that increased knowledge among pharmacists, coupled with guidelines and greater monitoring of the legislation, has appreciably reduced the purchasing of antibiotics without a prescription, providing guidance to key groups in Albania [43]. There are also concerns with current antibiotic prescribing patterns in hospitals in Albania. This includes their prolonged use post-operatively to prevent surgical site infections (SSIs) [44]. Previous published studies have also documented appreciable utilization of ‘Watch’ antibiotics in Albania compared with other European countries [45,46]. This is also a concern as high inappropriate use of ‘Watch’ antibiotics will drive up AMR [18,19].

Consequently, in view of ongoing concerns in Albania, we believe there is a need to build on previous publications that have tracked antibiotic utilization patterns in Albania, including high rates of ‘Watch’ antibiotics. [23,45,46]. This is important with delays in the approval of the Albanian NAP to reduce AMR, which was drafted in October 2017 [47]. The tracking of antibiotic utilization patterns can be used as a starting point for examining and debating future strategies to address utilization patterns of concern, such as excessive use of ‘Watch’ antibiotics, as seen in Estonia and Slovakia [24,26]. Alongside this, we examine whether issues, such as perceived levels of corruption or changes in the wealth of a country in terms of gross domestic product (GDP), as seen in Albania, appear to influence antibiotic use. Such findings can also form part of future strategies to reduce AMR in a country. Our findings, alongside exemplars from other European countries and beyond, can subsequently be used to suggest additional strategies that could be undertaken by key stakeholder groups in Albania to improve future antibiotic utilization patterns and reduce AMR if concerns still exist. Currently, no additional activities appear to have been undertaken in Albania in recent years to improve future appropriate antimicrobial utilization among patients in line with the goals of the NAP. This also needs to be addressed.

Alongside this, we are aware that there has been an appreciable increase in the use of antibiotics in patients with COVID-19 across countries and sectors despite limited evidence of bacterial infections or co-infections [48,49,50,51,52,53], which might represent significant levels of inappropriate use. Consequently, there is also a need to examine antibiotic utilization trends during the pandemic in Albania alongside the implications.

In view of this, the objectives of this study were to measure and analyze antibiotic utilization patterns in Albania in recent years alongside possible factors influencing these patterns. Potential factors include an ageing population, increasing GDP, as well as current levels of corruption. The findings, along with exemplars from other countries, can subsequently be used to give guidance to key stakeholder groups in Albania and beyond.

## 2. Results

We will first discuss the broader antibiotic utilization patterns in Albania during the past 11 years using a traditional drug utilization approach before looking critically at their composition, including trends in the utilization of ‘Watch’ antibiotics. In addition, we assess the possible impact of key factors, including an ageing population and governance, on utilization patterns. There is no attempt to look critically at the utilization of individual antibiotics because there is no diagnostic or indication data in the data sets used for analysis. In addition, the lack of specific demand-side measures at set times precludes the use of sophisticated statistical analyses, including time series analyses.

### 2.1. Utilization Patterns

Total antibiotic consumption fell in Albania between 2011 and 2019 but increased thereafter until 2021, which coincided with the COVID-19 pandemic (Figure 1). There was a similar pattern in the overall utilization of the top 10 antibiotics. Their percentage utilization accounted for 69% to 81% of total antibiotic utilization during the study period. However, there was no obvious overall prescribing pattern during this period in terms of their percentage of total utilization. This is different to the utilization pattern of ‘Watch’ antibiotics. Their utilization appreciably increased from 10% of total utilization among the top 10 most utilized antibiotics (DID based) in 2011 to 82% in 2021. This is further illustrated in Figure 1 and Figure 2.

Overall, in 2011 only two antibiotics from the ‘Watch’ list were present in the list of the top 10 most utilized antibiotics in 2011. This had appreciably changed by 2021 when only two of the antibiotics in the top 10 were from the ‘Access’ list; the remainder were from the ‘Watch’ list (Table 1). Table 1 documents the increasing utilization of azithromycin as well as levofloxacin and other ‘Watch’ antibiotics during the study period.

### 2.2. Potential Influencing Factors

Table 2 shows potential factors that could have contributed to increased or decreased antibiotic utilization in Albania between 2011 and 2021.

Table 3 shows the Kendall’s tau-b correlation coefficients between antibiotic use and other potential factors. A strong correlation between the median age of the population and antibiotic utilization until the start of the pandemic (2011–2020) was observed. This may reflect reduced requests for antibiotics to treat children with acute respiratory diseases during this period with ageing populations. Between 2011 and 2020, there was a significant correlation with GDP per capita, while no correlation was observed with the corruption index (Table 3).

There is a significant correlation between the decrease in antibiotic use and an increase in median age for the period 2011–2020. A similar significant correlation is observed between the decrease in antibiotic use and the growth in GDP per capita. It must be noted that these correlations are significant only when the 2021 values, which we considered outliers, are excluded (Table 3).

The scatter plots in Figure 3 illustrate a potential relationship between antibiotic consumption and median age of the Albanian population and GDP per capita, with the value for year 2021 considered an outlier. We do not see any potential relationship between antibiotic consumption and the corruption index score.

## 3. Discussion

We believe this is the first study undertaken in Albania in recent years to document current antibiotic consumption through an 11-year study as well as changes in utilization patterns prior to the introduction of the NAP. This includes changes in patterns following the recent COVID-19 pandemic. Prior to the pandemic, we saw a decreasing trend in the overall utilization of antibiotics in Albania from 27.2 DIDs in 2011 to 18.8 DIDs in 2019. We believe this is primarily due to ageing populations with less childhood respiratory illnesses. Such a correlation is supported by the fact that over 90% of antibiotic utilization globally is in ambulatory care, with only a minority in hospital care, especially in LMICs [16,55]. Alongside this, ageing populations will mean less prescribing of antibiotics for self-limiting upper respiratory tract infections (URTIs) in children, with URTIs currently accounting for an appreciable proportion of antibiotic utilization in ambulatory care in LMICs [16,56]. In addition, strengthening of the healthcare system coupled with improved living standards, alongside improved community infrastructures (Table 3), may also have contributed to less transmission of communicable diseases and consequently lower utilization of antibiotics [34,57]. This is because we are unaware of systematic educational and other activities in Albania in recent years to address concerns with variable knowledge of key healthcare professionals (HCPs) regarding antibiotics and AMR and the subsequent impact on patient and physician behavior [38,42]. The high empiric use of antibiotics in these studies reflects a lack of culture and sensitivity testing, which coupled with patient pressures on physicians and pharmacists to prescribe and dispense antibiotics [38,42], needs to be addressed to reduce AMR in the future.

These pressures, coupled with increasing concerns with AMR, may have also contributed to the increasing use of ‘Watch’ antibiotics in recent years. In 2011, only two antibiotics from the ‘Watch’ list were in the top 10 most utilized antibiotics in Albania (Table 1, Figure 2). However, by 2019, antibiotics in the ‘Watch’ list accounted for seven of the top ten antibiotics and 70% of total usage of the top ten antibiotics. This is appreciably higher than the 60% target for ‘Access’ antibiotics within total antibiotic utilization in countries established by the WHO [16,17,19,55]. We are unable to comment further on this, alongside any statistical analysis, without additional studies and more sophisticated datasets. In any event, this trend needs to be urgently addressed to reduce AMR rates in Albania.

It is difficult to also comment on trends in the utilization of antibiotics in hospitals in Albania during the study years, including IV versus oral use, as we were unable to disaggregate the utilization data into separate sectors and components. However, there have been concerns in previous studies with prolonged antibiotic administration post-surgery to reduce SSIs in Albania [44]. This may be driven by concerns with coverage and resistance rates, with the potential for greater prescribing of ‘Watch’ antibiotics if there is high empiric prescribing with currently limited culture and sensitivity testing in hospitals in Albania, which is similar to a number of other countries documented in recent studies and reviews among LMICs [58,59]. Again, we are unable to comment further without additional research. However, we are aware that antimicrobial stewardship programs (ASPs) have been successfully introduced in a number of LMICs to reduce prolonged prescribing of antibiotics post-operatively to prevent SSIs (Appendix A). This is welcomed as there have been concerns with the instigation of ASPs among LMICs due to resource issues both in terms of personnel as well as the necessary financial resources needed for their instigation [60]. The implementation of ASPs can also help rapidly progress patients from IV to oral equivalents to hasten discharge where pertinent [61,62]. This is because we have seen the number of IV packs purchased increasing in Albania in recent years, especially in particular during the pandemic years (I Hoxha personal communication). These various ASP studies can act as exemplars to key stakeholder groups involved with hospital prescribing of antibiotics in Albania going forward as they start to implement the NAP.

Another identified concern is the increased utilization of antibiotics following the COVID-19 pandemic, which reversed the downward utilization trend that was seen (Figure 1). Such an upward trend seemed to have overridden previous influencing factors, including ageing populations, GDP, and corruption. The reasons for this trend are uncertain. However, this may demonstrate the considerable focus on managing patients with COVID-19 in Albania since the start of the pandemic, with this trend matching the increasing use of antibiotics seen in other countries during the pandemic [48,50,51]. This trend is difficult to justify as there is little evidence of bacterial infections in these patients [51,63,64]. This signals appreciable inappropriate prescribing in Albania, which again urgently needs to be addressed.

The increasing use of ‘Watch’ antibiotics, now accounting for 82% of total antibiotic utilization among the top 10 most utilized antibiotics during the pandemic, is also a considerable concern. This excessive use may have been driven by physicians feeling more comfortable with prescribing ‘Watch’ antibiotics to feel safer in the absence of resistance data with limited testing in reality. However, again, we cannot comment further without additional research. In any event, this trend also needs to be urgently reversed. Appendix A provides examples of ASPs that have been successfully undertaken in LMICs to improve physician prescribing of antibiotics. These examples are again welcomed given earlier concerns with successfully instigating ASPs in LMICs [60], providing exemplars to key stakeholder groups in Albania at this critical time as they start to again implement their NAP. Appendix A also provides examples of successful programs introduced in LMICs to reduce the purchasing of antibiotics without a prescription, which has also been documented as a concern in Albania despite current legislation [38,41].

We are also likely to see a greater use of prescribing indicators across countries, including Albania, centered on the AWaRe book with their treatment suggestions for a number of infections routinely seen across sectors [16,55]. This is seen as a key way of reducing AMR [16,17,55]. Other key activities that could be undertaken in Albania to improve the use of antibiotics include improving the education of all healthcare students, especially physicians and pharmacists, regarding their appropriate use. The curriculum should include current guidelines and their rationale, such as the AWaRe book, alongside input into AMR and ASPs. Making sure physicians and pharmacists are aware of the likely origins of infections, such as URTIs and COVID-19, and the limited impact on patient care with antibiotics, should also be part of the curricula as well as the likely impact of their overuse, including adverse reactions and AMR. Educational input should also be provided post qualification through lifelong mandatory education. This is because we have seen educated pharmacists in LMICs be proactive with the use of guidelines to reduce unnecessary dispensing of antibiotics without a prescription, which includes patients with COVID-19, and this can be applied in Albania (Appendix A) [43,65,66,67]. These campaigns have worked in other LMICs providing direction to key stakeholders in Albania as they seek to implement the NAP as well campaigns among patients regarding antibiotics, viral infections, and AMR to reduce their requests for antibiotics among both physicians and community pharmacists [56]. This will be a growing area among LMICs, with most studies to date on patient education conducted among higher-income countries [56]. We will be following this up in future research projects building on this study for which the strength is a long follow-up period using standard drug utilization methodologies endorsed by the WHO [23,45,68,69]. Overall, to enhance the appropriate use of antibiotics in Albania, the MoH in Albania will need to firstly reinforce the monitoring of the prescribing and utilization of antibiotics across sectors as well as within the distribution chain as part of NAP activities. Secondly, the various institutions need to foster education and awareness regarding antibiotics, AMR, and ASPs with all HCP groups as well as with the patient population at large. Thirdly, the various institutions and the MoH need to provide the necessary equipment and tools to HCPs to increase prescription accuracy and protocol compliance in line with AWaRe guidance.

We are aware of a number of limitations with this study. These include the fact that we could not split the utilization data down into ambulatory and hospital sectors for comparison purposes. In addition, since the data were aggregated, we were unable to break this down into the different formulations. There were also no diagnostic or resistance data to assess the appropriateness of prescribing, especially the ‘Watch’ antibiotics. Alongside this, data sets post pandemic were limited to undertake sophisticated statistical analysis of any trends. Despite these limitations, we believe our findings are robust based on the length of this longitudinal study and breaking antibiotic utilization patterns down into the different AWaRe components. Alongside this, potential different hypothesis and factors should be tested to guide all key stakeholders in Albania going forward when the NAP will be launched.

## 4. Materials and Methods

Antibiotic utilization patterns for the total population of Albania from 2011 to 2021 are documented as DIDs (defined daily doses per 1000 inhabitants per day). DIDs are an internationally recognized metric used by the WHO for undertaking drug utilization studies to meaningfully document drug utilization patterns across countries, including European countries, where there are appreciable differences in population sizes [23,26,45,69]. Individual antibiotics are broken down by their anatomical therapeutic chemical (ATC) and AWaRe classifications for comparative purposes as well as providing targets for future ASPs [14,15,70]. The ATC/DDD index toolkit from 2011 to 2021 was used for each respective year to calculate consumption for comparative purposes. The ATC and DDD methodologies are internationally recognized for analyzing utilization trends both within and across countries [20,68,69,71,72].

The AWaRe classification is important given previously high rates of ‘Watch’ antibiotics being prescribed and dispensed in Albania [23]. In this system, the ‘Access’ group are typically considered as first- or second-line antibiotics for up to 26 common or severe infections, generally with a narrow spectrum and low resistance potential. The ‘Watch’ group have a higher resistance potential and greater side effects, with the ‘Reserve’ group only recommended as last resort for severe infections [14,15,17,55].

We also listed the number of antibiotics in the ‘Watch’ vs. ‘Access’ categories each year in the top 10 most utilized antibiotics since these are the most prescribed. We chose the top 10 as this usually encompasses most of the utilized antibiotics and gives a good focus for future ASPs. Examples of successful ASPs will also be documented to give guidance to all key stakeholder groups across the sectors in Albania to improve future antibiotic prescribing and dispensing where concerns have been identified. This though will not be a systematic review of successful examples in the three key situations, e.g., hospital antibiotic prescribing to prevent SSIs, initiatives to improve physician prescribing, as well as initiatives to appreciably reduce the purchasing of antibiotics without a prescription. This is because one of the primary objectives of this paper is to provide examples to key stakeholder groups in Albania regarding successful approaches that have been implemented in other LMICs to address concerns with utilization patterns where these are seen rather than undertake full systematic reviews. These examples from the literature are based on the considerable experiences of the co-authors publishing in this area [10,20,21,43,56,58,73]. These examples can subsequently be appraised by key stakeholder groups in Albania for their local suitability as the government and others in Albania seek to reduce AMR in line with the goals of the NAP. This builds on similar situations that some of the co-authors have been involved with across LMICs [10,56,58,73]. We concentrated on ASPs and their impact rather than any cost-effectiveness analyses of possible different interventions as we are aware that most of these have been undertaken in high income countries [74,75,76,77]. Consequently, the approaches may not be totally suitable for LMICs [60].

The source of the antibiotic utilization data was the import records provided from the Drug Agency and Customs in Albania combined with the sales records from local manufacturers. This encompasses the total population of Albania.

EUROSTAT with INSTAT (Albanian Institute for Statistics) provided population data [36], with the World Bank and Transparency International providing data on GDP as well as the corruption index [37,54]. GDP data is important, especially in countries with high co-payment levels for medicines as well as those where there is appreciable purchasing of antibiotics without a prescription as seen in Albania [38,41]. The corruption index is also important where regulations need to be enforced to reduce the inappropriate use of antibiotics, including their dispensing without a prescription [32,33]. The age profile of patients is also important with, as mentioned, over 90% of antibiotic utilization across countries, typically in ambulatory care coupled with considerable over prescribing and dispensing of antibiotics for essentially viral infections, including coughs and colds, among children [16,17,56,73].

The Kendall’s tau-b correlation coefficient with 95% confidence interval (two-tailed) was calculated to test potential associations between antibiotic utilization patterns and potential key factors using IBM SPSS Statistics (version 26). Two-sided *p*-values < 0.05 were considered statistically significant. Scatterplots were subsequently created using Microsoft Excel to illustrate potential associations.

There was no ethical approval for this study as no patients were involved. In addition, data was extracted from publicly available datasets. This is in line with institutional guidance.

## 5. Conclusions

In conclusion, there are principally three challenges to address in Albania to improve future antibiotic use given current concerns, especially with the growing use of ‘Watch’ antibiotics. Firstly, key stakeholders need to reinforce and implement the draft NAP leveraging multi-sectorial. Secondly, ASPs need to be urgently introduced in hospitals to address concerns with excessive prescribing of antibiotics. This includes any prolonged use to prevent SSIs, any prolonged IV administration, or excessive prescribing in patients with COVID-19. Thirdly, universities need to make sure in their curricula that all key HCP students are fully conversant with the rationale for prescribing antibiotics, especially ‘Watch’ antibiotics’, as well as undertaking ASPs post-graduation.

Lastly, the MoH in Albania will need to reinforce the routine monitoring of prescribing and utilization of antibiotics across sectors as well as provide the necessary equipment and tools to HCPs to increase prescription accuracy and protocol compliance in line with AWaRe guidance.

## Figures and Tables

**Figure 1 antibiotics-12-00882-f001:**
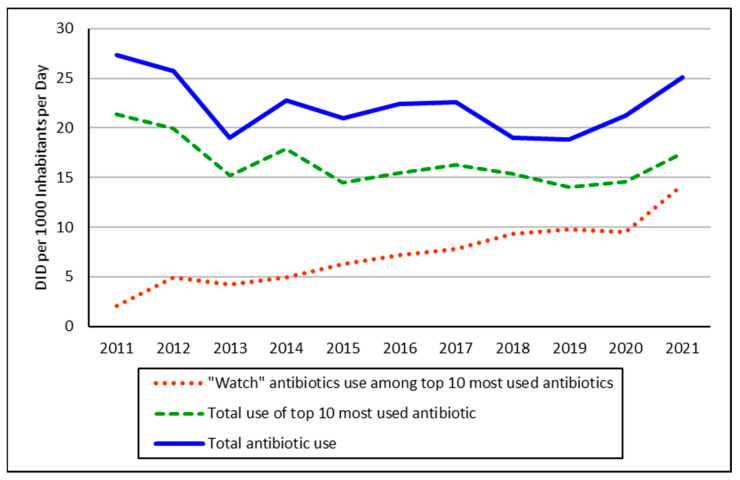
Trends in antibiotic utilization 2011–2021. NB: ‘Watch’ antibiotics based on the AWaRe classification [14,15]—See Section 4 for details.

**Figure 2 antibiotics-12-00882-f002:**
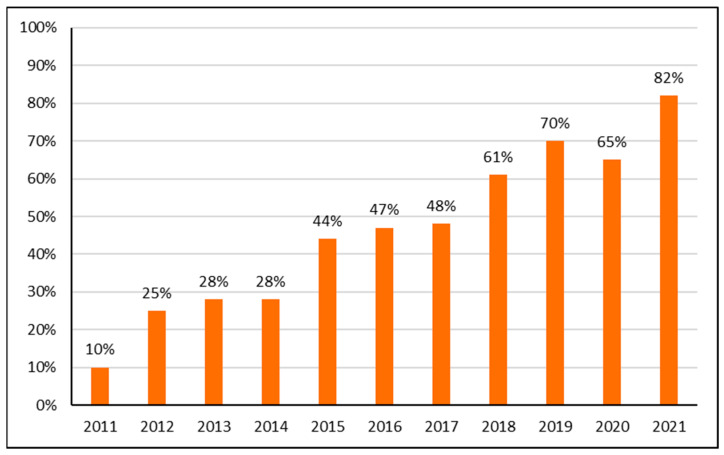
‘Watch’ antibiotics as percentage of the top 10 most utilized antibiotics 2011—2021.

**Figure 3 antibiotics-12-00882-f003:**
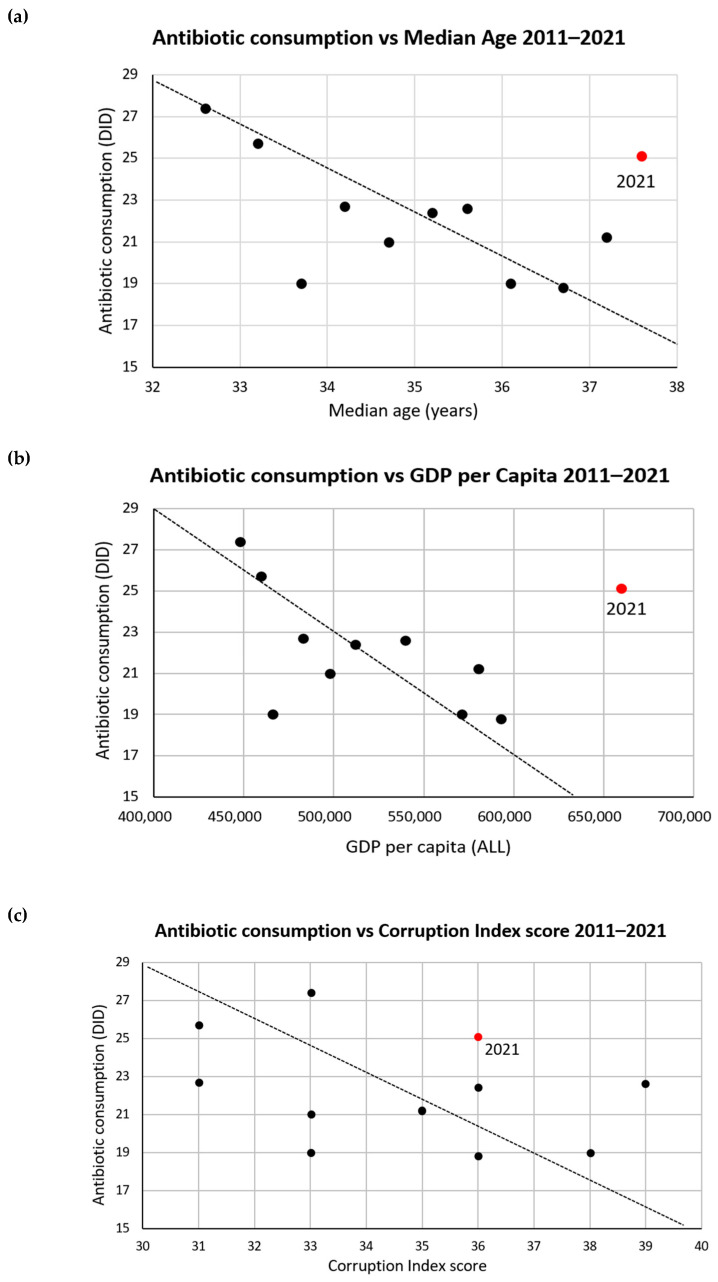
Scatterplots of antibiotic consumption vs. other factors in Albania. (**a**) Antibiotic consumption vs. median age of Albanian population 2011–2021. (**b**) Antibiotic consumption vs. GDP per capita in Albania 2011–2021. (**c**) Antibiotic consumption vs. corruption index score of Albania 2011–2021.

**Table 1 antibiotics-12-00882-t001:** Top 10 most used antibiotics in Albania expressed in DID from 2011 up to 2021.

2011	DID	2012	DID	2013	DID	2014	DID	2015	DID	2016	DID	2018	DID	2018	DID	2019	DID	2020	DID	2021	DID
Tetracycline	6.5	Amoxicillin	6.0	Amoxicillin	3.9	Amoxicillin	5.8	Amoxicillin	3.1	Co-amoxiclav	3.9	Co-amoxiclav	3.4	Co-amoxiclav	2.6	Co-amoxiclav	2.3	Azithromycin	3.0	Azithromycin	4.3
Amoxicillin	5.5	Co-amoxiclav	3.4	Co-amoxiclav	3.4	Co-amoxiclav	3.4	Ciprofloxacin	2.4	Cefuroxime	1.9	Amoxicillin	2.8	Cefuroxime	1.9	Cefuroxime	2.0	Co-amoxiclav	2.7	Levofloxacin	2.7
Co-amoxiclav	2.4	Metronidazole	3.2	Ciprofloxacin	1.7	Tetracycline	2.1	Co-amoxiclav	2.0	Amoxicillin	1.7	Cefuroxime	2.2	Amoxicillin	1.8	Azithromycin	1.9	Levofloxacin	1.4	Co-amoxiclav	2.1
Ciprofloxacin	1.3	Ciprofloxacin	1.9	Tetracycline	1.2	Ciprofloxacin	1.7	Tetracycline	1.6	Tetracycline	1.6	Ciprofloxacin	2.0	Ciprofloxacin	1.8	Cefaclor	1.5	Cefuroxime	1.4	Ciprofloxacin	1.5
Doxycycline	1.3	Moxifloxacin	1.2	Doxycycline	1.2	Doxycycline	1.1	Cefuroxime	1.3	Ciprofloxacin	1.4	Doxycycline	1.5	Tetracycline	1.6	Ciprofloxacin	1.2	Ceftriaxone	1.4	Cefixime	1.5
Cefazolin	1.1	Azithromycin	0.9	Azithromycin	1.0	Cefuroxime	1.0	Azithromycin	1.3	Azithromycin	1.4	Azithromycin	1.2	Cefaclor	1.5	Amoxicillin	1.1	Amoxicillin	1.2	Ceftriaxone	1.4
Cotrimoxazole	0.9	Cefaclor	0.9	Cefuroxime	0.8	Azithromycin	1.0	Cefixime	0.8	Clarithromycin	1.1	Ceftriaxone	0.9	Azithromycin	1.3	Levofloxacin	1.1	Tetracycline	1.2	Cefuroxime	1.2
Moxifloxacin	0.9	Cefalexin	0.8	Clarithromycin	0.7	Clarithromycin	0.8	Doxycycline	0.8	Doxycycline	0.9	Clarithromycin	0.8	Cefixime	1.0	Ceftriaxone	1.1	Ciprofloxacin	0.9	Tetracycline	1.0
Cefaclor	0.8	Cotrimoxazole	0.8	Ampicillin	0.7	Ampicillin	0.6	Clarithromycin	0.6	Ceftriaxone	0.8	Cefixime	0.8	Levofloxacin	1.0	Cefixime	1.0	Cefixime	0.8	Clarithromycin	0.9
Ampicillin	0.7	Doxycycline	0.7	Cotrimoxazole	0.5	Ceftriaxone	0.5	Nitrofurantoin	0.6	Levofloxacin	0.7	Nitrofurantoin	0.7	Ceftriaxone	0.8	Doxycycline	0.8	Cefaclor	0.7	Cefaclor	0.9
Total	21.4		19.9		15.2		17.9		14.5		15.4		16.3		15.3		14.0		14.6		17.5
Grand Total	27.4		25.7		19.0		22.7		21.0		22.4		22.6		19.0		18.8		21.2		25.1

NB: Antibiotics in the ‘Access’ list are in green, and those in the ‘Watch’ list are in Red. ‘Access’ and ‘Watch’ antibiotics are based on the AWaRe classification [14,15]. See Section 4 for details.

**Table 2 antibiotics-12-00882-t002:** Antibiotic utilization and other potential factors.

Year	Antibiotic Utilization (DIDs) *	Median Age (Years)	GDP per Capita (ALL) *	Albania Corruption Index Score *
2011	27.40	32.60	447,689	33
2012	25.70	33.20	459,526	31
2013	19.00	33.70	466,325	33
2014	22.70	34.20	482,954	31
2015	21.00	34.70	497,902	33
2016	22.40	35.20	511,971	36
2017	22.60	35.60	539,645	39
2018	19.00	36.10	571,011	38
2019	18.80	36.70	592,779	36
2020	21.20	37.20	580,521	35
2021	25.10	37.60	660,168	36

NB: * Data from [36,37,54]; DDDs per 1000 inhabitants per day; ALL = Albanian Lek.

**Table 3 antibiotics-12-00882-t003:** Correlations between antibiotic utilization and potential factors.

Years	Antibiotic Utilization (DIDs) VS:	Kendall’s Tau-b Correlation Coefficient	*p*-Value (2-Tailed)	95% Confidence Intervals (2-Tailed) ^a^	
				Lower	Upper
2011–2020	Median age (years)	−0.494 *	0.048	−0.756	−0.098
	GDP per capita (ALL)	−0.539 *	0.031	−0.781	−0.158
	Albania corruption index score	−0.262	0.312	−0.663	0.255
2011–2021	Median age (years)	−0.294	0.212	−0.660	0.185
	GDP per capita (ALL)	−0.330	0.160	−0.682	0.146
	Albania corruption index score	−0.216	0.378	−0.610	0.264

NB: DIDs = DDDs per 1000 inhabitants per day; ALL = Albanian Lek. ^a^ Estimation is based on Fisher’s r-to-z transformation. * Correlation is significant at the 0.05 level (2-tailed).

## Data Availability

Additional data are available on reasonable request from the co-authors. However, all sources have been referenced.

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
