# Peer review of "11-Year Trend in Antibiotic Consumption in a South-Eastern European Country; the Situation in Albania and the Implications for the Future"

_antibiotics, 2023, doi:10.3390/antibiotics12050882_

Round 1

Reviewer 1 Report

It is an interesting study aimed to assess the trend of antibiotic use in Albania from 2011-2021 based on AWaRe classifications. Hope my comments can improve the manuscript.

Minor comments:

1.       Please mentioned the objectives of the study clearly at the end of the introduction section.

2.       Line 259: Pearson?

3.       Table 2: Consider revising the appearance of the table so that the readers can easily understand the information.

4.       Line 259: Did the authors have checked the assumptions of 11 data points so that the use of Pearson correlation is valid? Is it normally distributed?

5.       Conclusions: Such statements should be placed in the discussion section. Please conclude based on the data presented and in line with the objectives of the study. Also mention the conclusion in the abstract.

Major comments:

1.       Line 245-247: The authors use “the sales records from local manufacturers” to capture the antibiotics utilization. How accurate is this data to capture the actual use in the population? Why did the authors not use sales records from hospitals/pharmacies/primary cares?

2.       Please describe the strength and limitations of this study in the discussions section.

Please check for any typos.

Author Response

Comments and Suggestions for Authors

  1. A) It is an interesting study aimed to assess the trend of antibiotic use in Albania from 2011-2021 based on AWaRe classifications. Hope my comments can improve the manuscript.

Author comments: Thank you for this and your suggestions. We hope we have adequately addressed these.

  1. B) Minor comments:
  2. Please mentioned the objectives of the study clearly at the end of the introduction section.

Author comments: Thank you – now added

  1. Line 259: Pearson?

Author comments: Thank you – now adjusted.

  1. Table 2: Consider revising the appearance of the table so that the readers can easily understand the information.

Author comments: Thank you for this comment. We have now updated Table 1. However, we would like to keep Table 2 as we believe this clearly shows the changes in the utilisation patterns with greater utilisation of Watch antibiotics in recent years. We also believe there is little we can add to Table 3, and Figure 1 represents Table 1. Hopefully, this is acceptable.

  1. Line 259: Did the authors have checked the assumptions of 11 data points so that the use of Pearson correlation is valid? Is it normally distributed?

Author comments: Thank you for this comments - yes already checked.

  1. Conclusions: Such statements should be placed in the discussion section. Please conclude based on the data presented and in line with the objectives of the study. Also mention the conclusion in the abstract.

Author comments: Thank you. We have now amended both to better link the conclusions with the Discussions, etc., and hope this is now OK.

  1. C) Major comments:
  2. Line 245-247: The authors use “the sales records from local manufacturers” to capture the antibiotics utilization. How accurate is this data to capture the actual use in the population? Why did the authors not use sales records from hospitals/pharmacies/primary cares?

Author comments: Thank you for this comment. We have used this methodology before in exercises with WHO Europe comparing utilisation patterns across Europe (which we cite), It is difficult to obtain individual data in Albania – hence we have utilised this approach over the years to gain a good understanding of the trends. We can subsequently look at utilisation patterns in individual hospitals and among physicians to enhance our knowledge especially when we undertake ASPs – the next steps. We hope this is now OK.

  1. Please describe the strength and limitations of this study in the discussions section.

Author comments: Thank you – now inserted.

Comments on the Quality of English Language. Please check for any typos.

Author comments: Thank you for this. We have now updated the manuscript with the help of one of the co-authors who is a native English speaker with over 500 publications in peer-reviewed Journals in recent years. We hope this is now acceptable.

Reviewer 2 Report

Thank you very much for the opportunity to review this article about the 11-year trend in antibiotic consumption in a South-Eastern Europe European country; the situation in Albania and the implications for the future.

Overall, I found it to be an average originality article, that offers a standard approach and, therefore, may not offer a valuable perspective for clinical practice.
Nevertheless, some aspects are amenable to revision. These are as follows:

In the materials and methods section (note that this should be reordered, as it should be placed before results), I propose that at the beginning, they describe the type of study being performed generally. Then, the eligible population and its characteristics. The latter, together with the selection criteria, is important to be well described since one of the main problems of this type of study lies precisely in the potential for selection bias.
The second section of the results (2.2. Potential influencing factors) would fit better in the discussion section.
The conclusions, in theory, derive entirely from the authors' research, so they should be triggered into one paragraph.

Author Response

Comments and Suggestions for Authors

1) Thank you very much for the opportunity to review this article about the 11-year trend in antibiotic consumption in a South-Eastern Europe European country; the situation in Albania and the implications for the future.

Overall, I found it to be an average originality article, that offers a standard approach and, therefore, may not offer a valuable perspective for clinical practice.

Author comments: Thank you for this. We do not believe such an in-depth review of current antibiotic utilisation patterns has been undertaken in Albania before – especially regarding key issues surrounding the use of Watch antibiotics in recent years. This is important as Albania seeks to launch its NAP – with this analysis providing key areas for the government and others to concentrate on to reduce AMR. Alongside this, possible reasons for an initial decline in view of key issues such as ageing populations. This is more in-depth than for instance multiple publications from the WHO European group published in the Lancet and Frontiers that principally document comparative utilisation patterns across countries with limited explanations (referenced) as well as earlier publications from the co-authors (Hoxha et al – documented) and those of e.g. Tesar et al documenting longitudinal data for Slovakia (cited). Subsequently, using the experiences of the co-authors as well as published papers to offer guidance in key areas with a number of exemplars across LMICs (documented as Supplementary Material). Potential guidance, etc., utilises the considerable experience of the co-authors who between them have published multiple studies discussing potential approaches to improve antibiotic utilisation to reduce AMR principally across LMICs. We hope this is now acceptable.

2) Nevertheless, some aspects are amenable to revision. These are as follows:

Author comments: Thank you for this – we hope we have adequately addressed your comments/ concerns

3) In the materials and methods section (note that this should be reordered, as it should be placed before results), I propose that at the beginning, they describe the type of study being performed generally. Then, the eligible population and its characteristics. The latter, together with the selection criteria, is important to be well described since one of the main problems of this type of study lies precisely in the potential for selection bias.

Author comments. Thank you for this. We have now made clearer that this is principally a drug utilisation study - building on previous studies documented by the WHO and the co-authors – and documents total antibiotic consumption in the country. Consequently, no selection bias as such, In addition, the order of the chapters is dictated by the Journal in their submission template. We hope this is OK with you.

4) The second section of the results (2.2. Potential influencing factors) would fit better in the discussion section.

Author comments: Thank you for this. May we beg to differ as we were looking at potential causes/ influences for the prescribing patterns seen based on key areas discussed in the Introduction. We then expanded on this in the Discussion. We hope this is OK with you.

5) The conclusions, in theory, derive entirely from the authors' research, so they should be triggered into one paragraph.

Author comments: Thank you for this comment. We have now updated the conclusion based on yours and other comments, and hope this is now acceptable.

Reviewer 3 Report

v    Strengths

The topic would be interesting as it deals with an emerging issue, antibiotic resistance and the proper use of antibiotics. The aim of the manuscript is to analyse trends in antibiotic consumption in Albania over the past 11 years.

The references cited are mostly recent (within the last 5 years) and relevant publications. 

Manuscript results are reproducible based on the details provided in the methods section. 

The results provide a slight advancement of current knowledge regarding antibiotic resistance in Albania.

v    Weaknesses

Too many self-citations.

The experimental design could be thorough and appropriate to test the hypothesis. The study is mostly descriptive, more statistical analysis of the data could be done to make the results even more interesting

Tables S1 and S2 of supplementary material are no references for the presented study; just put the quotations and comparisons (maybe not all) in the discussions.

Abstract:

line 20: better explain 'watch' antibiotics.

Line 25: what are the indicators?

line 26: explain the acronym “DIDs” as it is the first time encountered

lines 25-30: adjust the sentence to make it clearer

Introduction:

Better explain and expand the section on AWaRe classification

lines 65-68: make the sentence clearer

Line 68-69: Explain further and expand on corruption

Lines 71-75: the sentence is unclear; please explain

Lines 83-85: this statement could be moved into the discussions because, so far, we know nothing about the consumption of “Watch” antibiotics in Albania being part of the study's objective.

Line 86: Why “Would”?

Line 90: Why “were”?

Lines 90-97: this paragraph could be moved to the discussions.

Results:

This session could be expanded and described more comprehensively.

Line 111: what are these antibiotics?

Table 1 is unclear. Specify that the 10 most prescribed antibiotics and the 10 most prescribed antibiotics in the 'watch' group are listed in Table 2.

line 137: "To treat ..." insert reference or explain more clearly how this statement was derived because an ageing population cannot automatically tell me that fewer antibiotics are prescribed to treat children.

table 3 unclear. please extend explanation

Discussion:

line 158: where does this figure come from?

line 161-162: explain this sentence more clearly

lines 167-169: explain this sentence more clearly

line 170: what pressures?

line 186: explain the acronym being the first time it is found in the text

lines 187-192: this paragraph may be removed.

The paragraph from line 211 to line 225 could be resized

Better explain the study's limitations

Tables S1 and S2 of supplementary material are no references for the presented study; just put the quotations and comparisons (maybe not all of them) in the discussions.

Materials and Methods:

Does this part follow the results? Methods do normally anticipate the results.

Shat kind of study was done? 

The data say up to 2022 (line 230), but in the results, there are only up to 2021.

Conclusions:

lines 278-283: the paragraph could be shortened as it is somewhat repetitive

v    Suggestions to Author/s

I would suggest to the author to include the article 10.1038/s41598-022-24563-1 at line 184 of the discussion as a suggestion to use technology to try to reduce surgical infections and thus the use of antibiotics.

Moderate editing of English language

Author Response

Comments and Suggestions for Authors

v    Strengths

The topic would be interesting as it deals with an emerging issue, antibiotic resistance and the proper use of antibiotics. The aim of the manuscript is to analyse trends in antibiotic consumption in Albania over the past 11 years.

The references cited are mostly recent (within the last 5 years) and relevant publications. 

Manuscript results are reproducible based on the details provided in the methods section. 

The results provide a slight advancement of current knowledge regarding antibiotic resistance in Albania.

Author comments: Thank you for this – appreciated!

v    Weaknesses

1) Too many self-citations.

Author comments: Thank you for this. The papers by Hoxha et al and those with the WHO AMC help set the scene for the study. The comments on other CEE/ SEE countries also build on previous publications of the co-authors, e.g. B & H, Poland and Slovenia, to endorse the rationale, etc., for the paper. We have now expanded on this with e,g. studies from Estonia, Slovakia, etc. The papers in the Supplementary Material give examples among others for guidance and the papers cited in line 382 provide justification for our approach including a lack of ethical approval. We hope this is acceptable

2) The experimental design could be thorough and appropriate to test the hypothesis. The study is mostly descriptive, more statistical analysis of the data could be done to make the results even more interesting

Author comments: Thank you for this comment. As mentioned, this is typically a descriptive study documenting utilisation patterns across multiple years. As such, builds on previous studies by the co-authors as well as the WHO AMC group previously published in the Lancet and Frontiers in Pharmacology (cited) and we believe the first study of its kind in a SEE country that documents utilisation patterns over such a long time period (11 years). Since there have been no major initiatives in the country to improve antibiotic utilisation – it is difficult to perform e.g. time series analyses (which the authors have performed in multiple publications), In addition – too few time points to fully analyse utilisation trends post COVID-19 and the lack of diagnostic and other data to fully interpret changes in the prescribing of Watch antibiotics. However – we have used statistical techniques to assess whether changes in key metrics such as ageing populations, improved infrastructure, etc., have made a difference. In addition, offered guidance to the key stakeholder groups across Albania on potential ways forward with ASPs, etc., to improve future patterns as Albania seeks to implement its NAP based on a comprehensive list of ASPs that have been successfully undertaken across sectors in LMICs as there have been concerns whether LMICs have the necessary capacity to perform ASPs. We hope this is now acceptable.

3) Tables S1 and S2 of supplementary material are no references for the presented study; just put the quotations and comparisons (maybe not all) in the discussions.

Author comments: Thank you for this. As explained in the original Discussion (where we quoted these Tables) we were looking to provide guidance to key stakeholders in Albania on ways to improve antibiotic utilisation in hospitals and ambulatory care given identified concerns especially in recent years and the forthcoming launch of the NAP. We have now developed this further to provide additional guidance to key stakeholder groups including ways to reduce unnecessary purchasing of antibiotics without a prescription. As such, we believe this consolidated approach will also be of interest to other LMIC in similar positions based on our combined experience. We trust this is now acceptable. 

A) Abstract:

1) line 20: better explain 'watch' antibiotics.

Author comments: Thank you - now done.

2) Line 25: what are the indicators?

Author comments: Thank you - now amended

3) line 26: explain the acronym “DIDs” as it is the first time encountered

Author comments: Thank you - now amended

4) lines 25-30: adjust the sentence to make it clearer.

Author comments: Thank you now amended.

B) Introduction:

1) Better explain and expand the section on AWaRe classification

Author comments: Thank you – now updated.

2) lines 65-68: make the sentence clearer

Author comments: Thank you – now amended. We hope this is now OK.

3) Line 68-69: Explain further and expand on corruption

Author comments: Thank you – now updated.  

4) Lines 71-75: the sentence is unclear; please explain

Author comments: Thank you – now updated.

5) Lines 83-85: this statement could be moved into the discussions because, so far, we know nothing about the consumption of “Watch” antibiotics in Albania being part of the study's objective.

Author comments: Thank you for this. However, as quoted, previous publications from the WHO have shown high use of Watch antibiotics in Albania – and we wanted to build on this given concerns with rising AMR rates globally and the implications. We have made this clearer, and hope this is now acceptable.

6) Line 86: Why “Would”?

Author comments: Thank you – as mentioned – we would like to build on previous publications that give concerns with antibiotic prescribing in Albania especially with concerns surrounding AMR and the launch of the NAP. We hope this is now clearer with the update.

7) Line 90: Why “were”?

Author comments: Thank you – changed to ‘are’

Lines 90-97: this paragraph could be moved to the discussions.

Author comments: Thank you for this. However, may we beg to differ as we wanted to set the scene why this study was important for key groups in Albania as key stakeholders seek to launch the NAP, etc., in an effort to improve future antibiotic utilization. We hope this is now OK.

C) Results:

This session could be expanded and described more comprehensively.

Line 111: what are these antibiotics?

Table 1 is unclear. Specify that the 10 most prescribed antibiotics and the 10 most prescribed antibiotics in the 'watch' group are listed in Table 2.

line 137: "To treat ..." insert reference or explain more clearly how this statement was derived because an ageing population cannot automatically tell me that fewer antibiotics are prescribed to treat children.

table 3 unclear. please extend explanation

Author comments: Line 111: Now amended to read total utilisation. Table 1 and 2 – now made the link clearer (we have also expanded the methodology section explaining why we have concentrated on the top 10. Line 137 – thank you for this. As explained in the methodology – most antibiotics are utilized in ambulatory care (typically over 90% of total consumption in LMICs) – and most is for self-limiting conditions such as URTIs in children. Consequently – we would expect to see overall lower utilisation of antibiotics if less children to treat inappropriately with antibiotics for their URTIs (lines 339 – 342). We hope this is now clearer. Table 3 just documents possible key influencers of antibiotic utilisation as discussed in the Introduction and Methodology. We hope this is now acceptable.

D) Discussion:

1) line 158: where does this figure come from?

Author comments: Thank you – we have referenced this from e.g. the AWaRe book and other cited publications. We hope this is acceptable.

2) line 161-162: explain this sentence more clearly.

Author comments: Now updated.

3) lines 167-169: explain this sentence more clearly

Author comments: Thank you – now updated.

4) line 170: what pressures?

Author comments: We know from multiple publications (cited) that patients can put considerable pressure on physicians to prescribe antibiotics, and pharmacists to dispense them, for e.g. URTIs for themselves/ their children unless appropriately educated (discussed further in some of the ASPs). We hope this is now acceptable.

line 186: explain the acronym being the first time it is found in the text

Author comments: Thank you – ASPs are a well know term to all key stakeholders involved in antibiotic use/ research. We have given multiple examples in the Supplementary Material, and hope this is now acceptable.

5) lines 187-192: this paragraph may be removed.

Author comments: Thank you – we beg to differ as we were looking to offer guidance to all key stakeholder groups in Albania and wider on potential ways to improve future utilization given concerns. We hope this is OK with you as we believe this is a key part of the paper going forward especially as Albania seeks to launch its NAP to reduce AMR.

6) The paragraph from line 211 to line 225 could be resized

Author comments: Thank you for this. Again as above we believe this is a key area going forward which we have further expanded in the Supplementary Materials. We hope this is also acceptable. 

7) Better explain the study's limitations

Author comments: Now done

8) Tables S1 and S2 of supplementary material are no references for the presented study; just put the quotations and comparisons (maybe not all of them) in the discussions.

Author comments: As discussed above we strongly beg to differ on this point especially as Albania seeks to launch its NAP and hope this is OK with you.

E) Materials and Methods:

1) Does this part follow the results? Methods do normally anticipate the results.

Author comments: Thank you – this is the format of the template supplied by the Journal. We hope this is acceptable

2) What kind of study was done? 

Author comment: Thank you – this is a typical drug utilization study following previous publications in the Lancet and Frontiers in Pharmacology (WHO AMC Network), We hope this is OK with you.

3) The data say up to 2022 (line 230), but in the results, there are only up to 2021.

Author comments: Thank you – now changed.

F) Conclusions:

lines 278-283: the paragraph could be shortened as it is somewhat repetitive

Author comments: Thank you – updated with the help of other.

v    Suggestions to Author/s

I would suggest to the author to include the article 10.1038/s41598-022-24563-1 at line 184 of the discussion as a suggestion to use technology to try to reduce surgical infections and thus the use of antibiotics.

Author comments: Thank you. We have included successful ASPs to reduce SSIs in the Supplementary material to give guidance (as documented above) and hope this is acceptable as this is the main aim of this section

Reviewer 4 Report

Good study. However the statistical analysis can be made better.

In the abstract, what does DID stand for? It should be spelt out before using it for the first time.

The authors have used Anatomical Therapeutic Classification (ATC) classification to classify antibiotics. However factors like route( whether Oral or injectable) have been ignored. Is there some way how it could be figured out if they are patented or off patent? Is there some reason how their cost per unit could be available.

Performing a Pearson's correlation with just 10 variables will not give the correct value. 

If you can get the median age and cost for each antibiotic for each year, I would suggest running a mixed model/ GEE with DID as the outcome, cost for each drug, AWARE classification category, median age for each drug in each year as predictors.

If not, with Table 2, you can run an empty mixed model with each antibiotic as it's own cluster. 

Author Response

Comments and Suggestions for Authors

1) Good study. However the statistical analysis can be made better.

Author comments: Thank you for this comment. As mentioned, this is typically a descriptive study documenting utilisation patterns across multiple years. As such, builds on previous studies by the co-authors as well as the WHO AMC group previously published in the Lancet and Frontiers in Pharmacology (cited) and we believe the first study of its kind in a SEE country that documents utilisation patterns over such a long time period (11 years). Since there have been no major initiatives in the country to improve antibiotic utilisation – it is difficult to perform e.g. time series analyses (which the authors have performed in multiple publications), In addition – too few time points to fully analyse utilisation trends post COVID-19 and the lack of diagnostic and other data to fully interpret changes in the prescribing of Watch antibiotics. However – we have used statistical techniques to assess whether changes in key metrics such as ageing populations, improved infrastructure, etc., have made a difference. In addition, offered guidance to the key stakeholder groups across Albania on potential ways forward with ASPs, etc., to improve future patterns as Albania seeks to implement its NAP based on a comprehensive list of ASPs that have been successfully undertaken across sectors in LMICs as there have been concerns whether LMICs have the necessary capacity to perform ASPs. We hope this is now acceptable

2) In the abstract, what does DID stand for? It should be spelt out before using it for the first time.

Author comments: Thank you – now addressed

3) The authors have used Anatomical Therapeutic Classification (ATC) classification to classify antibiotics. However factors like route( whether Oral or injectable) have been ignored. Is there some way how it could be figured out if they are patented or off patent? Is there some reason how their cost per unit could be available

Author comments: Thank you for these comments. Since our focus was on total utilisation patterns – we just documented these as difficult to break down this data into the different formulations when documenting total DIDs. In addition – the main reason for looking at IV vs. oral is in hospital to ensure a more rapid conversion to oral therapy to reduce length of stay and costs (which we mention). As this is also a traditional drug utilisation study (building on previous studies in the Lancet and Frontiers in Pharmacology with the WHO AMC group) – costs do not come into it – this is a separate study. We hope this is now clear.  

4) Performing a Pearson's correlation with just 10 variables will not give the correct value. 

If you can get the median age and cost for each antibiotic for each year, I would suggest running a mixed model/ GEE with DID as the outcome, cost for each drug, AWARE classification category, median age for each drug in each year as predictors.

Author comments: Thank you for this comment. We believe this is fine. Since this is a drug utilisation study and we did not look at e.g. costs, and with no diagnostic data to evaluate the appropriateness of the prescribing of Watch antibiotics, there is no value or role in performing more sophisticated analyses than those undertaken, The same philosophy with undertaking any time series analysis without target health policy interventions to concentrate on. The intention of this study was to analyse current patterns and possible causes where we can to highlight areas for future interventions (with examples) as the NAP is launched. We hope this is now acceptable.

5) If not, with Table 2, you can run an empty mixed model with each antibiotic as it's own cluster. 

Author comments: Thank you. As discussed, this is not appropriate without the diagnostic data. All we can do (as performed) is to show increasing concerns with appreciable prescribing of Watch antibiotics and the implications. This can be taken further in specific ASPs for each sector/ group. This is why we spent considerable time and effort with the examples in the Supplementary Material. We hope this is now acceptable.

Round 2

Reviewer 1 Report

The discussion section would be a better home for the authors' arguments and suggested future paths, which make up the majority of the conclusion section.

Author Response

Comments and Suggestions for Authors

The discussion section would be a better home for the authors' arguments and suggested future paths, which make up the majority of the conclusion section.

Author comments: Thank you – now changed. We hope this is now acceptable.

Reviewer 2 Report

This is OK with me

Author Response

Comments and Suggestions for Authors

This is OK with me

Author comments: Thank you for this – appreciated!

Reviewer 3 Report

Thank you for your comments, they have made the manuscript and the study project clearer.

I think some points that were not changed as suggested could make it more valuable. 

The self-citations, although you have explained why they are there, I still consider them excessive.

Regarding Tables S1 and S2, as much as you explained your intent to provide guidance to key stakeholders in Albania on ways to improve antibiotic utilisation in hospitals and ambulatory care, from the title of your manuscript and the results of your study, it appears more of an analysis, description and commentary regarding antibiotic consumption. The tables could be used in case to do a systematic review with the aim of giving guidance to key stakeholders in Albania. 

Author Response

Comments and Suggestions for Authors

1) Thank you for your comments, they have made the manuscript and the study project clearer.

I think some points that were not changed as suggested could make it more valuable. 

Author comments: Thank you – we have addressed these where we can. Please though bear in mind that drug utilisation studies such as these are a documentation of current utilisation patterns, with the findings used to suggest potential ways forward where concerns are identified (as in this case). We have tried - as seen - to look at a number of potential explanations for the utilisation patterns that we saw with the limited data sets we have, i.e. no diagnostic data, etc. This is in line with the many drug utilisation studies we and others have undertaken across disease areas, countries and medicines including those orchestrated by the WHO in Europe (we have been part of this network) – as e.g. emphasized in lines 103 and 104 in the revised paper. We hope this is now acceptable.

2) The self-citations, although you have explained why they are there, I still consider them excessive.

Author comments: Thank you – we have further removed some of these (as indicated) – especially regarding the lack of ethical approval – which you have accepted. However, we have kept important ones that document ongoing activities and their impact in Central and Eastern European countries (e.g. Azerbaijan, Poland, Republic of Srpska and Slovenia) as well as the current situation in Albania (Hoxha et al). In addition, examples among pharmacies in Africa (Kenya and Namibia) as well as general reviews regarding the appropriate use of antibiotics in LMICs and NAPs in Africa (background and examples). We have also added in a number of additional references from other authors as examples, etc., to further help in this area. We hope this is now OK.

3) Regarding Tables S1 and S2, as much as you explained your intent to provide guidance to key stakeholders in Albania on ways to improve antibiotic utilisation in hospitals and ambulatory care, from the title of your manuscript and the results of your study, it appears more of an analysis, description and commentary regarding antibiotic consumption.

Author comment: Thank you for this – this is the aim of drug utilisation studies – especially in the absence of crucial diagnostic and other data.

4) The tables could be used in case to do a systematic review with the aim of giving guidance to key stakeholders in Albania. 

Author comments. Thank you for this. We have now provided additional guidance/ rationale of our methodology/ approach, etc., in the updated methodology. As you are no doubt aware, to perform updated systematic reviews in this area will need to be part of extensive MSc/ PhD projects, which is well outside the scope of this paper. We have though provided additional examples for guidance, etc., and trust this is now acceptable.

Reviewer 4 Report

Dear authors:

The issue of antibiotic resistance in Middle Income Countries is an important topic and happy to learn that research in this field is being undertaken. May I suggest the following:

Line 45: There should be a space between 4.95 and the word million.

Line 83: The word "population" is twice in the same sentence. Please rewrite.

Line 102: The word "this" has to be added after Thus. 

Line 277: The line ends abruptly saying regarding their appropriate. 

I have concerns regarding Table 3 and use of Pearson's correlation coefficient. 

Consider using non-parametric alternatives, such as Spearman's rank correlation or Kendall's tau, which are less sensitive to outliers and do not assume a linear relationship between variables. These methods might be more appropriate for a small sample size.

Include a confidence interval for the correlation coefficient or a p-value to help assess the significance of the relationship between the variables. This will provide readers with more information on the uncertainty of the results, given the small sample size.

Include a scatterplot of the data to provide a visual representation of the relationship between the variables. This can help readers to better understand the nature of the relationship, including linearity, outliers, and the presence of potential confounding variables.

See comments above. 

Author Response

Comments and Suggestions for Authors

Dear authors:

The issue of antibiotic resistance in Middle Income Countries is an important topic and happy to learn that research in this field is being undertaken.

Author comments: Thank you for these kind comments – appreciated.

May I suggest the following:

1) Line 45: There should be a space between 4.95 and the word million

Author comments: Thank you – now changed.

2) Line 83: The word "population" is twice in the same sentence. Please rewrite.

Author comments: Thank you – now refined.

3) Line 102: The word "this" has to be added after Thus. 

Author comments: Thank you – now changed.

4) Line 277: The line ends abruptly saying regarding their appropriate. 

Author comments: Thank you – now updated. We hope this is now OK.

5) I have concerns regarding Table 3 and use of Pearson's correlation coefficient. 

Consider using non-parametric alternatives, such as Spearman's rank correlation or Kendall's tau, which are less sensitive to outliers and do not assume a linear relationship between variables. These methods might be more appropriate for a small sample size.

Include a confidence interval for the correlation coefficient or a p-value to help assess the significance of the relationship between the variables. This will provide readers with more information on the uncertainty of the results, given the small sample size. Include a scatterplot of the data to provide a visual representation of the relationship between the variables. This can help readers to better understand the nature of the relationship, including linearity, outliers, and the presence of potential confounding variables.

Author comments: Thank you for this guidance. As seen, we have now updated the analysis using Kendall's tau as well as performed scatter plots. This includes excluding 2021 from some analyses as an outlier. We hope this is now acceptable.
